# Is a Virtual Reality Test Able to Predict Current and Retrospective ADHD Symptoms in Adulthood and Adolescence?

**DOI:** 10.3390/brainsci9100274

**Published:** 2019-10-13

**Authors:** Débora Areces, Trinidad García, Marisol Cueli, Celestino Rodríguez

**Affiliations:** 1Department of Psychology, University of Oviedo, 33003 Asturias, Spain; arecesdebora@uniovi.es (D.A.); cuelimarisol@uniovi.es (M.C.); rodriguezcelestino@uniovi.es (C.R.); 2Faculty Padre Ossó of Education, University of Oviedo, 33003 Asturias, Spain

**Keywords:** virtual reality, ADHD, attention, adults, adolescents, assessment

## Abstract

Despite the persistence of attention deficit hyperactivity disorder (ADHD) into adulthood and adolescence, there are few objective, reliable instruments (based on patient performance) that have been shown to be able to predict current and retrospective ADHD symptoms. The present study aimed to explore whether a validated VR test called Nesplora Aquarium is able to predict ADHD symptoms in adults and adolescents, based on both current and retrospective self-reports. A non-clinical sample of 156 adults and adolescents (70 women and 86 men) between 16 and 54 years of age (*M* = 21.23, *SD* = 8.04) took part in the study. Virtual reality (VR) variables such as the number of correct answers, omission and commission errors, among others, were used to predict current and retrospective self-reported symptoms of ADHD using multiple regression models. Correct answers and omission errors in the VR test significantly predicted both current and retrospective ADHD symptoms. However, only the number of perseveration errors and gender were able to significantly predict retrospective ADHD symptoms. These findings suggest that inattention problems tend to remain after adolescence, while perseveration errors (which have been related to impulsive behavior) and gender differences tend to diminish.

## 1. Introduction

Attention deficit hyperactivity disorder (ADHD) is a neurodevelopmental disorder characterized by persistent attentional problems, difficulty in controlling impulses, and hyperactivity [1]. It affects between 5% and 13% of children [2,3], interfering in their school and family adaptation. After adolescence, impulsivity symptoms tend to diminish [4], but it is estimated that ADHD continues to affect between 40% and 70% of those cases [5], and that it continues to interfere in social and labor adaptation [6,7]. A previous study showed that people who reported having ADHD symptoms in childhood not only generally had a lower educational level, but were also more likely to be unemployed, abuse drugs, or be imprisoned [8].

Previous neuroimaging studies suggested that people with ADHD show reduced activation, compared to controls, in right lateral prefrontal cortex regions when performing go/no go tasks. This suggests that a deficit in executive function (EF) gives rise to the behavioral symptoms of ADHD [9]. EF includes functions such as monitoring and actualizing information in working memory, inhibiting undesired responses or avoiding paying attention to irrelevant stimuli, and shifting attention between activities [10,11]. There are a wide variety of continuous performance tests (CPT) aimed at children and adolescents that have been shown to be reliable, valid instruments in the diagnosis of ADHD. These tests provide quantitative data on different variables of interest that have been shown to be related to ADHD symptoms [12,13,14]. The most common variables provided by CPT are the following:

Correct responses: the sum of the participant’s correct responses, considered as a measure of working memory due to the high cognitive load involved in responding simultaneously in the two sensory modalities with varying instructions.

Omission errors: the sum of the participant´s errors, failing to press the button when they were supposed to press it, considered an indicator of attentional arousal in situations of high cognitive load for working memory.

Commission errors: the sum of the participant´s errors, pressing the button when they were not supposed to press it, considered an indicator of impulsivity.

Reaction time: the average time the participant took to press the button once the target stimuli was presented, considered an indicator of processing speed.

Variability of reaction time: the difference between the fastest reaction time and slowest reaction time registered in the test. This measure is indicative of changes in sustained attention or fatigability during the task.

Conversely, in the case of the adult population, various studies have noted the existence of underdiagnoses of ADHD. The main reason is due to the absence of CPTs or other objective tools (based on patient performance) which allow assessment of possible ADHD cases in adolescence and more especially in adulthood. Both late adolescence and adulthood require more complex testing and assessment protocols because of the presence of comorbid psychiatric disorders.

Considering the complexity of diagnosing ADHD in adulthood or late adolescence, it is important to have adequate objective tests. Most current tests demonstrate poor ecological validity and are administered in unrealistic conditions that might generate extraneous variance depending on the many ways subjects adapt to these conditions [15]. Another more frequently used option are observation scales, which might be more sensitive to how EFs are used in real life [9], although they might be biased by the respondents’ subjectivity [15,16].

It is important to develop reliable, ecologically valid measures for measuring ADHD symptoms and the incorporation of virtual reality (VR) is a promising approach [15]. While maintaining the objectivity of typical performance tests, VR tests reduce the extraneous variance associated with the artificiality of the test conditions because participants are immersed in a realistic environment. Recent studies showed that VR tests of EF were useful for exploring the differences between children with ADHD and controls, as well as in different ADHD presentations [16]. However, in adults, although different EF VR tests have been validated [17], their effectiveness in predicting ADHD symptoms has not yet been established.

This study aims to explore whether a validated VR test called Nesplora Aquarium [18] is able to predict ADHD symptoms in adults and adolescents, based on both current and retrospective self-reports. Nesplora Aquarium is a VR-CPT designed to measure EF in adolescents and adults. Participants wear three-dimensional glasses equipped with sensors and headphones and are “immersed” in a virtual aquarium. Then, following the go/no-go paradigm, they perform the following tasks: (1) Learning task training 1/Learning task 1: This task consists of an AX (go task) or 1-back paradigm; the button must be pressed whenever the person sees a clownfish or hears the word “clownfish”, if the previous fish or word has been a surgeon. (2) Task 1 (Dual execution-Xno training/Dual execution-Xno task): This is a Dual X_no or Dual No_go task. The person must press the button whenever a fish appears or a word is heard except when seeing the Clownfish or hearing the word “surgeon”. (3) Task 2 (Dual execution-Inversion of the target stimuli in Dual execution-Xno task): This is a Dual X_no or Dual No_go task. The person must press the button whenever a fish appears or a word is heard except when seeing the surgeon or hearing the word “clownfish”. In this sense, through the inversion of the target stimuli it is possible to evaluate the control of interference by both switching capacity (cost of task change) and perseveration errors.

Each of these tasks is composed of 140 evaluation items and 20 training items, but only the evaluation items of the final two tasks are considered in the analysis. The objective of the first task is to produce fatigue and ensure learning of the stimuli. The whole test takes about 20 min. Once the tasks are finished, the new VR test not only provides the aforementioned variables (correct responses, omissions, commissions, response time, and variability), but also provides data on two new important variables: (1) switching: the difference between the participant´s correct responses in the last part of a task and the beginning of the next task, considered an indicator of the capacity to shift attention; and (2) perseveration errors: the errors registered in the dual execution tasks, when subjects respond to the task by following the instruction of the previous task. This variable is interpreted as deficit in cognitive flexibility.

Given the potential benefits of using VR tools that provide different variables related to attention, impulsivity, and hyperactivity problems, it is likely that VR variables will be significant predictors of current and retrospective ADHD symptoms.

This objective would have significant practical implications in the diagnosis of ADHD in adolescence and adulthood. If a VR test was able to predict a high percentage of retrospective patient symptoms, VR variables could be used as a good indicator for determining whether ADHD symptomatology had been present throughout a patient’s life.

## 2. Materials and Methods

### 2.1. Sample

A non-clinical sample of 156 adults took part in the study: 70 women (44.9%) and 86 men (55.1%), between 16 and 54 years of age (*M* = 21.23, *SD* = 8.04). The sample was recruited by a convenience method, which means that all participants were previously informed about the objectives of the present research. The participants came from different high schools, universities, and companies in Spain. Participants reported a medium-high socio-economic level. Regarding education, 13.4% had studied up to secondary school level and 86.6% had university qualifications. From the complete sample, 53.3% were in mid and late adolescence (between 16 and 20 years of age, *M* = 16.74, *SD* = 1.097), while the remaining 46.8% were adults (between 21 and 54 years of age, *M* = 27.55, *SD* = 9.108). None of the participants reported having been diagnosed with ADHD or other psychiatric or neurological diseases.

### 2.2. Procedures

The study had the approval of the pertinent Ethical Committee of the Principality of Asturias (reference: CPMP/ICH/135/95, code: TDAH-Oviedo), and all procedures were performed in compliance with relevant laws and institutional guidelines. After providing informed consent, participants completed two self-report measures (current ADHD symptoms and retrospective ADHD symptoms) and completed the Nesplora Aquarium VR test. The evaluations were conducted in a laboratory and lasted for 1 h. A member of the research group was responsible for contacting the participants and supervising the assessment process.

### 2.3. Instruments

The Spanish adaptation [19] of the Adult ADHD Self-Report Scale (ASRS) [20] was used to measure current ADHD symptoms. This consists of six items where participants report the presence of inattention symptoms (four items) or hyperactivity symptoms (two items) in the previous 6 months, using a 5-point Likert scale ranging from 0 (never) to 4 (very often). Various authors have stated that this scale is useful not only for assessing current ADHD in adults, but also in adolescents [21].

A cut-off score ≥ 12 suggests that symptoms are consistent with ADHD (sensitivity = 96.7%, specificity = 91.1%). Ramos-Quiroga et al. [19] reported moderate internal reliability (α = 0.72), while in the present study it was lower (α = 0.61).

To measure retrospective ADHD symptoms, the ADHD subscale from the Wender Utah Rating Scale (WURS) [22] was administered to the participants, using the Spanish adaptation [23]. It consists of 25 items that rate how frequently different ADHD symptoms were present during childhood, using a 5-point Likert scale (from 0 = not at all or very slightly to 4 = very much). A cut-off score ≥ 32 suggests that symptoms are consistent with ADHD in childhood (sensitivity = 91.5%; specificity = 90.8%). The internal reliability of the scale was high, both in the validation study (α = 0.94) [20] and in the present study (α = 0.89).

Finally, the VR EF Test called Nesplora Aquarium [18] (described in detail above) was used. The normative study from this new virtual reality test demonstrated alpha coefficient values for the two dual tasks of 0.975 (task 2 or Dual execution 0-Xno task) and 0.968 (task 3 or Inversion of the target stimuli in Dual execution-Xno task).

### 2.4. Data Analyses

Different analyses were performed. First, descriptive statistics (mean, standard deviation, asymmetry, and kurtosis) were calculated (Table 1).

Then, two hierarchical regression analyses were conducted (Table 2): one analysis to predict current symptoms (ASRS) and the other to predict retrospective symptoms (ADHD-WURS). Taking into consideration the wide age range of the sample, this variable was included as a possible predictor variable in the regression analyses. Thus, the first of the two different regression models tested included the effect of age group (group 1: 16–20; group 2: 21–25; group 3: 26–30; group 4: 31–35; group 5: 36–40; group 6: 41–45; group 7: 46–50; and group 8: 51–55) and gender as well as the Nesplora Aquarium general measures (number of correct answers, omissions, commissions, response time, variability, switching, and perseveration errors). The second model added the number of omissions, commissions, and response time obtained in task 1; and finally, model 3 added the omissions, commissions, and response time in task 2 to the previous model. The selection of these variables was based on previous studies which have used CPTs [14,16,17]. Preliminary analyses based on normal P-P plots and scatterplots of the standardized residuals were conducted to ensure that the data met requirements of normality, linearity, homoscedasticity, and independence of residuals, following the criteria from Tabachnick and Fidell [24]. Analyses of VIF (Variance Inflation Factor) coefficients were performed to ensure no violation of the assumption of multicollinearity (VIF < 10). All statistical analyses were conducted with SPSS v24.0 (Chicago, IL, USA) [25], with a significance level of *p* < 0.05.

## 3. Results

Table 2 shows the results of the hierarchical regression analyses. These results indicate that model 1 (which incorporates the effect of gender, age group, and general variables of Nesplora Aquarium) predicted the majority of variance explained in current ADHD symptoms. Models 2 and 3 (which incorporated variables related to task 1 or task 2) led to increases in the variance explained which were not significant. More specifically, out of all of the variables included in model 1, only the number of correct answers and omission errors were statistically significant variables in the prediction of ADHD symptoms.

Similarly, with the retrospective symptoms, model 1, with four significant predictors, explained 88.80% of the variance in childhood ADHD symptoms, and the introduction of other variables in models 2 and 3 also showed no significant increases in the variance explained.

## 4. Discussion

This study supports the utility of VR EF measures for predicting current and retrospective ADHD symptoms in adolescents and adults. Specifically, the number of correct responses in the test and the number of omission errors were associated with current and retrospective ADHD symptoms, while gender and the number of perseveration errors were only related to retrospective ADHD symptomatology. These findings suggest that the number of omissions and correct answers are the only variables able to predict both current and retrospective ADHD problems. The findings suggest that inattention problems tend to continue after adolescence, while perseveration errors (which have been related to impulsive behavior) and gender differences tend to diminish [4,12,13,14,16].

The associations found between the VR measures and ADHD symptoms were higher than those found when traditional EF performance measures were used [26]. In particular, this tool exhibited a great advantage because it was able to predict 88.80% of retrospective ADHD symptoms. It gives clinicians the chance to complement subjective information about patients’ childhoods with objective VR measures. This result shows that this VR test provides reliable and significant predictors in the diagnosis of ADHD in adolescent and adults, since it was designed to eliminate problems of respondent subjectivity and maintain ecological validity, and at the same time provide a variety of objective measures in ADHD diagnosis. Despite these findings, some limitations in the present study must be acknowledged. First, the small sample size may limit the generalizability of the results. Particularly, it would be beneficial to add a clinical sample in order to verify the differences between ADHD group in comparison to the control group in the present variables analyzed. Also, the wide age range of the sample (comprised of adolescents and adults) must be considered. Although the effect of age group was examined in the different regression models, the fact that ADHD symptoms may vary over time may necessitate a concern regarding the generalization of our results, according to the approach given to this study (cross-sectional) and the limited number of participants in each age sample. Second, additional measures of ADHD symptoms, such as the DSM-5 protocol [1], could be included in future research to give a better picture of the symptoms. Last, it is necessary to highlight the limitations due to the scale used for current ADHD symptoms (comprising six items) and the scale used for retrospective ADHD symptoms (comprising 25 items).

## 5. Conclusions

In summary, Nesplora Aquarium has been shown to provide significant predictors of current and retrospective ADHD symptoms in adults and adolescents, combining independence from respondent subjectivity and ecological validity. However, continued work on that explanation is crucial to determine how the symptoms can be either accentuated or masked in adulthood, their real implications in daily life, and the design of appropriate interventions. It is especially important if we consider the potential high prevalence of ADHD symptoms in adulthood [26].

## Figures and Tables

**Table 1 brainsci-09-00274-t001:** Descriptive statistics for current symptoms (ASRS), retrospective symptoms (attention deficit hyperactivity disorder Wender Utah Rating Scale (ADHD-WURS)), and virtual reality (VR) performance measures.

Test	Variables	*M*	*SD*	Skewness	Kurtosis
ASRS	Total score	8.71	3.76	0.424	0.242
WURS	ADHD subscale	22.28	12.83	1.012	2.621
VR performance measures Nesplora Aquarium	Number of correct answers	356.58	41.44	−1.218	1.264
	Number of omissions	37.74	37.28	1.908	3.318
	Number of commissions	25.68	13.69	1.348	2.847
	Response time (msec)	881.17	65.53	0.039	1.074
	Variability	601.61	127.71	−0.078	−0.147
	Switching	1.69	4.66	0.707	1.077
	Perseveration errors	16.78	7.61	0.329	−0.243

Note: *M* = mean; *SD* = standard deviation.

**Table 2 brainsci-09-00274-t002:** Hierarchical regression analysis models to predict ADHD symptoms at present (ASRS) and retrospectively (ADHD-WURS).

	Variables	ASRS	ADHD-WURS
Model 1	Gender β (*t*)	0.039 (1.384)	0.100 (2.603 *)
	Age group β (*t*)	−0.060 (−1.799)	−0.039 (−0.845)
	Number of correct answers β (*t*)	0.983 (3.919 ***)	0.859 (2.504 *)
	Number of omissions β (*t*)	0.159 (3.253 ***)	0.204 (3.066 ***)
	Number of commissions β (*t*)	0.072 (1.263)	−0.020 (−0.252)
	Response Time β (*t*)	−0.244 (−0.731)	0.054 (0.117)
	Variability β (*t*)	0.016 (0.114)	−0.352 (−1.820)
	Switching β (*t*)	−0.16 (−0.637)	−0.012 (−0.356)
	Perseveration errors β (*t*)	0.074 (0.934)	0.229 (2.117 *)
	*R* ^2^	0.940 ***	0.888 ***
Model 2	Gender: β (*t*)	0.031 (1.087)	0.104 (2.651 **)
	Age group β (*t*)	−0.068 (−2.035 *)	−0.039 (−0.846)
	Number of correct answers β (*t*)	0.839 (3.235 ***)	0.800 (2.241 *)
	Number of omissions β (*t*)	0.078 (0.912)	0.045 (0.383)
	Number of commissions β (*t*)	0.014 (0.221)	−0.018 (−0.203)
	Response Time β (*t*)	−0.069 (−0.112)	−0.045 (-0.053)
	Variability β (*t*)	−0.036 (−0.244)	−0.308 (−1.523)
	Switching β (*t*)	0.001 (0.041)	0.028 (0.670)
	Perseveration errors β (*t*)	0.045 (0.566)	0.208 (1.889)
	Omissions Task1 β (*t*)	0.066 (0.922)	0.160 (1.630)
	Commissions Task 1 β (*t*)	0.096 (1.927)	0.002 (0.029)
	Response Time Task 1 β (*t*)	0.039 (0.074)	0.130 (0.179)
	*R* ^2^	0.942 ***	0.890 ***
	Δ*R*^2^	0.002	0.002
Model 3	Gender β (*t*)	0.031 (1.069)	0.105 (2.639 **)
	Age group β (*t*)	−0.059 (−1.725)	−0.032 (−0.681)
	Number of correct answers β (*t*)	0.998 (3.598 ***)	0.958 (2.493 *)
	Number of omissions β (*t*)	−0.021 (−0.179)	−0.045 (−0.276)
	Number of commissions β (*t*)	0.042 (0.633)	0.010 (0.104)
	Response Time β (*t*)	0.437 (0.581)	0.379 (0.364)
	Variability β (*t*)	0.042 (0.262)	−0.250 (−1.125)
	Switching β (*t*)	−0.028 (−0.804)	0.000 (0.001)
	Perseveration errors β (*t*)	0.123 (0.509)	0.354 (1.055)
	Omissions Task 1 β (*t*)	0.092 (1.252)	0.182 (1.787)
	Commissions Task 1 β (*t*)	0.081 (1.574)	−0.009 (−0.132)
	Response Time Task 1 β (*t*)	−0.300 (−0.517)	−0.180 (−0.224)
	Omissions Task 2 β (*t*)	0.054 (0.520)	0.033 (0.231)
	Commissions Task 2 β (*t*)	−0.076 (−0.417)	−0.130 (−0.511)
	Response Time Task 2 β (*t*)	−0.395 (−1.218)	−0.330 (−0.734))
	*R* ^2^	0.943 ***	0.891 ***
	Δ*R*^2^	0.001	0.001

Note: β = standardized beta coefficient; *t* = Student *t* coefficient; *R*^2^ = variance explained; Δ*R*^2^ = change in variance explained. * *p* < 0.05; ** *p* < 0.01; ***, *p* < 0.001.

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
