# Peer review of "Is a Virtual Reality Test Able to Predict Current and Retrospective ADHD Symptoms in Adulthood and Adolescence?"

_brainsci, 2019, doi:10.3390/brainsci9100274_

Round 1

Reviewer 1 Report

The authors propose to use their newly developed VR test for the assessment of attention and working memory capacity (Nesplora Aquarium) to predict current and retrospective ADHD symptoms. Their results suggest that the number of omissions predicts both retrospective and current ADHD symptoms, whereas the number of commissions only predicts retrospective ADHD symptoms. According to the authors, these findings would suggest that inattention problems of ADHD patients do not decrease from adolescence to adulthood, whereas impulsivity symptoms tend to diminish.

This is a concise paper, with a straightforward rationale. I think that this new VR tool and these results could be of great interest for many researchers and clinicians working with ADHD patients. I however have some major concerns and suggestions below, as well as some specific comments, that I hope are useful.

1. The main objective of this study is to predict current and retrospective ADHD symptoms in adulthood. In the introduction, the authors emphasize the differences of ADHD symptoms between adolescence and adulthood. But then, we see in the methods that participants between 16 and 54 years old have been tested by the authors. Aren’t 16-year-old participants adolescents? How could “retrospective” and “current” ADHD symptoms of these young participants be assessed, as they do not have reached adulthood yet? I am surprised that the authors did not have some inclusion criteria regarding the age of the participants. How many of the participants are actually really into adulthood? Also, I would be curious to see the results without these young participants, maybe by separating participants into age groups (e.g., adolescents, young adults and adults).

2. Another issue concerns the specificity of the dependant variables from the VR test, and especially of the number of correct answers and of the Switching. 1/ To me, there are two types of correct answers: correct go and correct no go. I believe that they might differently predict ADHD symptoms. One could for example expect that correct no-go responses would be a better indicator of inhibitory control than correct go responses, and would more efficiently predict ADHD symptoms. I would therefore suggest to add these different response types as separate variables to the model. 2/ In its current form, the “switching” variable corresponds to the difference in number of correct responses between the ending of a given task and the beginning of the subsequent task. But I would rather have assessed, in a same task, the difference in reaction times between go trials preceded by a no-go trial and go trials preceded by a go-trial. I wonder if this measure would have been more sensitive to attentional flexibility capacity, and would maybe have been a more efficient predictor of ADHD symptoms.

3. I am a bit surprised that the current-ADHD-symptoms scale used by the authors is only composed of 6 items, whereas the scale for retrospective symptoms is composed of 25 items. The main results concern the fact that some dependant variables (e.g., commission errors) predict the ADHD symptoms assessed with one scale but not the ones assessed with the other scale. I believe this type of results could simply reflect differences in sensitivity or internal reliability between the two scales, instead of actual differences between retrospective and current ADHD symptoms. If the authors do not have any possibility to control for this issue, I would at least recommend broaching this limitation into the discussion.

4. As far as I understand, the original article about the Nesplora Aquarium VR test provides more dependant variables (11 I think) than the ones assessed in the current paper (7). Why the authors did not include all the initially described dependant variables in their model, and how did they choose the ones to assess in the current study?

Specific comments:

- L.38: could the authors be more specific about the frontal regions involved in the cognitive deficits of ADHD patients? Also, some references to neuroscientific studies would be needed here.

- L.39-40: the sentence gives the impression that EF allow paying attention to irrelevant stimuli. On the contrary, I believe the authors wanted to highlight that EF allow avoiding to pay attention to irrelevant stimuli.

- L.44: I think that the variables of interest should be defined here. With the current form of the manuscript, the reader has to wait until the methods section to finally understand what the authors are referring to here. Same thing for the VR test, it should be described in the introduction, around L.65.

- Moreover, the description of the VR test (being L.65 or in the methods) should be more detailed. E.g., the tasks should be fully described in this article, instead of having to read the authors’ previous work to understand the VR tasks.

- L.49: I have troubles to understand what the “which” is referring to.

- L.76: I don’t understand what the authors mean by “the sample was recruited by convenience methods”.

- L.129-130: a lot of dependant variables are introduced here, whereas only some of them have been defined earlier in the paper. For example, it is never explained how variability and perseveration errors have been calculated.

- Table 1: Does “asymmetry” refer to the distribution skewness? If so, I think it should be renamed as such.

- L.140 and L.144, the authors say that Nesplora Aquarium variables predict 92% of the variance for current ADHD symptoms and 88% of the variance for retrospective ADHD symptoms. But I am not sure this is true, as the model also includes age and gender effects. Although age and gender do not significantly predict ADHD symptoms, the variance explained by these variables should be taken into account when reporting the variance explained by the VR test dependant variables.

- Table 2: although the perseveration errors seem to significantly predict retrospective ADHD symptoms, this is never mentioned in the text by the authors, and it is not discussed either. Is there a reason for that? It could be interesting to discuss this result as well, as it would suggest that preservative behaviors in ADHD tend to diminish in adulthood.

- L.158-161: I think that this reasoning could also be applied to the number of correct responses that predict current, but not retrospective ADHD symptoms. This is actually where the separate analysis of correct-go responses and correct-no-go responses could lead to an interesting discussion.

Author Response

The author acknowledge the comments from the reviewer and the possibility to submit a new version of the manuscript. The changes made in the manuscript, in accordance with the reviewer comments, are listed below. Moderate English changes were also made.  

Aren’t 16-year-old participants adolescents?

- We have modified the title of the present brief report, and clarified that the participants who took part in this study belonged to mid-late adolescence and adulthood.

How many of the participants are actually really into adulthood?

-We have provided that information in the sample section (L160-161).

Another issue concerns the specificity of the dependant variables from the VR test, and especially of the number of correct answers and of the Switching. 1/ To me, there are two types of correct answers: correct go and correct no go. 2/ In its current form, the “switching” variable corresponds to the difference in number of correct responses between the ending of a given task and the beginning of the subsequent task. But I would rather have assessed, in a same task, the difference in reaction times between go trials preceded by a no-go trial and go trials preceded by a go-trial. I wonder if this measure would have been more sensitive to attentional flexibility capacity, and would maybe have been a more efficient predictor of ADHD symptoms.

-Regarding to the first point, we have included the comment from the reviewer as a future research line, because when we will get a representative sample, we will introduce in the regression model more specific variables (i.e., omissions in go task Vs omissions in no go tasks) (you can see the comment in the discussion section). The present results from general variables analyzed allow to compare them with other studies which have used other CPT (the majority of them not based on VR) and provided similar measures.

On the other hand, the analysis proposed for the “switching” variable is very interesting, but at the moment is not possible, since the creators only provide us the switching variable in the current form described in the test.

I am a bit surprised that the current-ADHD-symptoms scale used by the authors is only composed of 6 items, whereas the scale for retrospective symptoms is composed of 25 items. If the authors do not have any possibility to control for this issue, I would at least recommend broaching this limitation into the discussion.

- As the reviewer suggested, we have added this suggestion as a limitation of the study in the discussion section (L321-325).

Why the authors did not include all the initially described dependant variables in their model, and how did they choose the ones to assess in the current study?

-We have selected the present variables based on previous studies with other Continuous Performance Test. In this sense, using the variables which are in common with other CPT not based on VR allows compare the results Aquarium Nesplora with other similar test.

Specific comments:

- L.38: could the authors be more specific about the frontal regions involved in the cognitive deficits of ADHD patients? -We have added the information requested (L40-41).

- L.39-40: the sentence gives the impression that EF allow paying attention to irrelevant stimuli. On the contrary, I believe the authors wanted to highlight that EF allow avoiding to pay attention to irrelevant stimuli. -We have corrected the mistake (L44).

- L.44: I think that the variables of interest should be defined here. With the current form of the manuscript, the reader has to wait until the methods section to finally understand what the authors are referring to here. Same thing for the VR test, it should be described in the introduction, around L.65. Moreover, the description of the VR test (being L.65 or in the methods) should be more detailed. E.g., the tasks should be fully described in this article, instead of having to read the authors’ previous work to understand the VR tasks. -We have reorganized the information as suggested by the reviewer. Moreover, we have also included a more detailed description of Nesplora Aquarium tasks (L101-109).

- L.49: I have troubles to understand what the “which” is referring to. - We have rephrased the paragraph to make it easier for readers to understand (L77-79).

- L.76: I don’t understand what the authors mean by “the sample was recruited by convenience methods”. -We have clarified the meaning of convenience methods (sample section, L157-158).

- L.129-130: a lot of dependent variables are introduced here, whereas only some of them have been defined earlier in the paper. For example, it is never explained how variability and perseveration errors have been calculated. - We have added that information (L141-144).

- Table 1: Does “asymmetry” refer to the distribution skewness?- We have modified the term “asymmetry” and renamed as “skewness”.

- L.140 and L.144, the authors say that Nesplora Aquarium variables predict 92% of the variance for current ADHD symptoms and 88% of the variance for retrospective ADHD symptoms. But I am not sure this is true, as the model also includes age and gender effects. Although age and gender do not significantly predict ADHD symptoms, the variance explained by these variables should be taken into account when reporting the variance explained by the VR test dependant variables. -We have included age and gender in the model in order to control the effect of these variables. In addition, as you can see in the regression model, the variables referred to gender and age were no statistically significant.

- Table 2: although the perseveration errors seem to significantly predict retrospective ADHD symptoms, It could be interesting to discuss this result as well, as it would suggest that preservative behaviors in ADHD tend to diminish in adulthood. -we have discussed the finding throughout the discussion section. Moreover, we have also mentioned the significance of perseveration errors in the result section.

- L.158-161: I think that this reasoning could also be applied to the number of correct responses that predict current, but not retrospective ADHD symptoms. This is actually where the separate analysis of correct-go responses and correct-no-go responses could lead to an interesting discussion. -we have added this comment as a limitation of the study.

Reviewer 2 Report

Manuscript ID: brainsci-573372

Title: Is a virtual reality test able to predict current and retrospective ADHD symptoms in adulthood?

Journal: Brain Sciences

Abstract

Page 1, line 16. Authors should add the number of males and females as well as the mean and standard deviation of age of the overall sample.

Introduction

Page 1, line 31. Authors wrote that ADHD “affects between 5% and 7% of children”. However, recent reports have shown a higher prevalence of this disorder in childhood (e.g., https://www.ncbi.nlm.nih.gov/pubmed/30646132). Authors wrote that CPT are useful for the diagnosis of ADHD in children (page 1, lines 41-43). As regards adults, authors wrote that “various studies have noted the existence of underdiagnoses of ADHD, mainly due to the lack of objective tools based on patient performance” (page 2, lines 47-48). I am wondering why CPTs are not commonly used in adults. Page 2, line 64. Authors should explicit the meaning of “differential utility”.

Materials and Methods

From my point of view, authors should necessary enrol a sample of ADHD patients aiming to ascertain the usefulness of their test in the discrimination between patients and controls. Page 2, line 82. Authors should add the name of the ethics committee who approved the study, as well as the report number of approval. Page 3, line 102. Authors quoted a normative study on this tool from their group, which is currently under review. Are there not any published data on the validity and reliability of their test? Page 3, lines 102-111. Aiming to facilitate the comprehension of their test, authors should describe an example of trial.

Results

Page 4, lines 140-141. Authors wrote that “Nesplora Aquarium variables significantly predicted 92.3% of the variance in current ADHD”. I would say that the model with two significant predictors (which were two out of seven Nesplora Aquarium variables inserted in the model) explained the 92.3% of the variance. The same comment also applies to description of the results referred to the retrospective symptoms (page 4, lines 143-145).

Discussion

Page 5, line 166. Authors wrote that “This result shows the advantages of using a VR test in the diagnosis of ADHD in adults”. I believe that this conclusion is not supported by the data presented by authors because ADHD patients were not examined.

Author Response

The author acknowledge the comments from the reviewer and the possibility to submit a new version of the manuscript. The changes made in the manuscript, in accordance with the reviewer comments, are listed below. Moderate English changes were also made.

Abstract: Page 1, line 16. Authors should add the number of males and females as well as the mean and standard deviation of age of the overall sample.

-We have added the information requested (see line 17-18).

Introduction. Page 1, line 31. Authors wrote that ADHD “affects between 5% and 7% of children”. However, recent reports have shown a higher prevalence of this disorder in childhood (e.g., https://www.ncbi.nlm.nih.gov/pubmed/30646132).

-We have modified the prevalence percentages and cited the mentioned study (L34).

As regards adults, authors wrote that “various studies have noted the existence of underdiagnoses of ADHD, mainly due to the lack of objective tools based on patient performance” (page 2, lines 47-48). I am wondering why CPTs are not commonly used in adults.

– At the moment clinicians do not have CPT addressed to assess cases from mid-late adolescence and adulthood (L77-80).

Page 2, line 64. Authors should explicit the meaning of “differential utility”.

-We have deleted the term differential, since the main objective consist on analyzing the utility of Nesplora Aquarium for predicting current and retrospective ADHD symptoms.

Materials and Methods: Page 2, line 82. Authors should add the name of the ethics committee who approved the study, as well as the report number of approval.

-We have incorporated the reference and code of the Ethical Committee of the Principality of Asturias (L165-166).

Page 3, line 102. Authors quoted a normative study on this tool from their group, which is currently under review. Are there not any published data on the validity and reliability of their test?

-We have added information about reliability of the test (L198-201).

 Page 3, lines 102-111. Aiming to facilitate the comprehension of their test, authors should describe an example of trial.

-We have increased the information related to the Nesplora Aquarium tasks (L101-L146).

Results: Page 4, lines 140-141. Authors wrote that “Nesplora Aquarium variables significantly predicted 92.3% of the variance in current ADHD”. I would say that the model with two significant predictors (which were two out of seven Nesplora Aquarium variables inserted in the model) explained the 92.3% of the variance. The same comment also applies to description of the results referred to the retrospective symptoms (page 4, lines 143-145).

-We have modified the paragraph following the recommendations from the reviewer (L219-222).

Discussion: Page 5, line 166. Authors wrote that “This result shows the advantages of using a VR test in the diagnosis of ADHD in adults”. I believe that this conclusion is not supported by the data presented by authors because ADHD patients were not examined.

-We have changed this expression according to the suggestions from the reviewer (L338-339).

Round 2

Reviewer 1 Report

In my previous review, I have raised several major concerns, and I am afraid that none of them has been addressed by the authors.

1. The inclusion of both adolescents and adults in the sample leads to several issues. First, it divides the sample size by two, as half of the participants are adolescents. Regarding this point, it would have been nice to indicate the cut-off between these two groups, and to provide the mean age for each group. Second, the authors use Adult scales to assess ADHD symptomatology, that does not seem to be suitable for assessing ADHD symptomatology in adolescents. It is surprising that the number of commission errors does not vary between adolescents and adults. The number of commission errors should predict adolescent current symptomatology. I believe this might be due to a poor estimation of ADHD symptomatology.

2. I have asked the authors to try some new analyses, but none of them has been performed during the revision. Moreover, their response in the point-by-point letter raises some more critical issues. 1/ The authors say that one of the analyses cannot be performed because “the creators only provide us the switching variable in the current form described in the test”. But this test has been created by the authors themselves. I therefore don’t understand why they couldn’t have access to the data. 2/ more critically, in their response the authors say that they will do one of the analyses in a future paper, while testing a more representative sample. Then, as a reader I would be more interested in that future paper. I am therefore uncertain about the potential impact of the current paper.

3. The selection of only 7 out of the 11 variables provided by the VR test has not been justified in the paper.

Author Response

The authors acknowledge the reviewers their suggestions and comments on the manuscript, and hope the changes we introduced will meet their expectations. These changes are described below, in response to the different aspects addressed in the revision. The authors have been reviewer editing problems and mistakes as well as edit grammar expressions for native speaker English.

Reviewer 1

Comments and Suggestions for Authors

The inclusion of both adolescents and adults in the sample leads to several issues. First, it divides the sample size by two, as half of the participants are adolescents. Regarding this point, it would have been nice to indicate the cut-off between these two groups, and to provide the mean age for each group. Second, the authors use Adult scales to assess ADHD symptomatology, that does not seem to be suitable for assessing ADHD symptomatology in adolescents. It is surprising that the number of commission errors does not vary between adolescents and adults. The number of commission errors should predict adolescent current symptomatology. I believe this might be due to a poor estimation of ADHD symptomatology. Considering these recommendations, we have added the mean and standard deviation for the adolescents and adults which composed the sample (see L124-125). Moreover, we have controlled the effect of age diving the sample in different age groups (see L160-161) and including the new variable called “the group of age” in the new hierarchical regression analysis, which have been conducted for checking the possible influence in the current and retrospective ADHD symptoms (see the changes in the results section). Regarding to the usefulness of ASRS scale in the assessment of ADHD symptoms in adolescents, many author have showed that this scale is appropriate not only for evaluating adults but only for adolescents (see L137-138).

I have asked the authors to try some new analyses, but none of them has been performed during the revision. Moreover, their response in the point-by-point letter raises some more critical issues.

1/ The authors say that one of the analyses cannot be performed because “the creators only provide us the switching variable in the current form described in the test”. But this test has been created by the authors themselves. I therefore don’t understand why they couldn’t have access to the data.

We are not the creators of this test. The tool was created by a private company called Nesplora which collaborates with our research team (specialized on assessment and intervention of ADHD symptoms) in order to carry out some studies. In this sense, we maintain daily contact in order to make recommendations and suggestions which allow to improve the quality of their services, but we can not modify directly the software of the test. Considering that, the system has allowed us to include some of the additional variables that you have proposed differentiating on the type of task (Omissions, commissions and response time task 1 Vs task 2) in the new analysis conducted. However, due to the limitations of the system, three variables (more specifically: switching, perseveration errors and variability) have to be incorporated as a general way (without differentiating between the type of the task) (see results section).

2/ more critically, in their response the authors say that they will do one of the analyses in a future paper, while testing a more representative sample. Then, as a reader I would be more interested in that future paper. I am therefore uncertain about the potential impact of the current paper.

We have analyzed the data considering your interesting suggestions. In this sense, two hierarchical regression models (one for the prediction of current ADHD symptoms and the other for the prediction of the retrospective ones) were calculated in order to verify whether the incorporation of new variables (omissions, commissions and response time obtained in task 1 Vs. omissions, commissions and response time in task 2) in the model 2 and 3 suppose a significant increase in the variance explained (L200-225).

The selection of only 7 out of the 11 variables provided by the VR test has not been justified in the paper. We have added the explanation about why we have chosen the present variables (see L164-167).

Reviewer 2 Report

Manuscript ID: brainsci-573372-peer-review-v2

Title: Is a virtual reality test able to predict current and retrospective ADHD symptoms in adulthood and adolescence?

Journal: Brain Sciences

Materials and Methods

It seems to me that the authors’ reply to the following comment of the first review is missing: “From my point of view, authors should necessary enrol a sample of ADHD patients aiming to ascertain the usefulness of their test in the discrimination between patients and controls”.

Discussion

Page 6, lines 223-224. I believe that authors should avoid any references to the usefulness of VR test in the diagnosis of ADHD because they did not enrol a clinical sample.

Author Response

The authors acknowledge the reviewers their suggestions and comments on the manuscript, and hope the changes we introduced will meet their expectations. These changes are described below, in response to the different aspects addressed in the revision. The authors have been reviewer editing problems and mistakes as well as edit grammar expressions for native speaker English.

Reviewer 2

Materials and Methods: It seems to me that the authors’ reply to the following comment of the first review is missing: “From my point of view, authors should necessary enrol a sample of ADHD patients aiming to ascertain the usefulness of their test in the discrimination between patients and controls”. Based on you interesting comment, we have modified paragraphs of the article in order to clarify the real objective of the present study which consist on exploring whether a validated VR test called Nesplora Aquarium is able to predict ADHD symptoms in adults and adolescents, based on both current and retrospective self-reports (see abstract and introduction section). Discussion: page 6, lines 223-224. I believe that authors should avoid any references to the usefulness of VR test in the diagnosis of ADHD because they did not enrol a clinical sample. We have modified the section and we have incorporated the necessity of including a clinical sample in order to analyzed the differences between ADHD group Vs. Control group (see L240-247).